# Interactions between Filter-Feeding Bivalves and Toxic Diatoms: Influence on the Feeding Behavior of *Crassostrea gigas* and *Pecten maximus* and on Toxin Production by *Pseudo-nitzschia*

**DOI:** 10.3390/toxins13080577

**Published:** 2021-08-19

**Authors:** Aurore Sauvey, Françoise Denis, Hélène Hégaret, Bertrand Le Roy, Christophe Lelong, Orianne Jolly, Marie Pavie, Juliette Fauchot

**Affiliations:** 1Normandie Université, UNICAEN, CNRS UMR 8067, BOREA, 14000 Caen, France; bertrand.leroy@unicaen.fr; 2Laboratoire de Biologie des Organismes et Ecosystèmes Aquatiques (BOREA)—Université de Caen Normandie, MNHN, SU, UA, CNRS UMR 8067, IRD 207, 14000 Caen, France; mariepavie@hotmail.com; 3Laboratoire de Biologie des Organismes et Ecosystèmes Aquatiques (BOREA)—MNHN, CNRS UMR 8067, SU, IRD 207, UCN, UA, Station de Biologie Marine, MNHN, 29900 Concarneau, France; francoise.denis@mnhn.fr; 4Laboratoire Mer, Molécules, Santé, EA 2160 MMS, Le Mans Université, CEDEX 9, 72085 Le Mans, France; 5Laboratoire des Sciences de l’Environnement Marin (LEMAR), UMR 6539 CNRS, UBO, IRD, Ifremer, IUEM, Technopôle Brest-Iroise, Rue Dumont d’Urville, 29280 Plouzané, France; Helene.Hegaret@univ-brest.fr; 6Normandie Université, UNICAEN, EA2608, OeReCa, 14000 Caen, France; christophe.lelong@unicaen.fr; 7Normandie Université, UNICAEN, Centre de Recherches en Environnement Côtier (CREC)—Station Marine, Université de Caen Normandie, 14530 Luc-sur-Mer, France; orianne.jolly@unicaen.fr

**Keywords:** domoic acid, filter-feeding bivalves, *Pseudo-nitzschia*, interactions, filtration, toxin accumulation, *Crassostrea gigas*, *Pecten maximus*

## Abstract

Among *Pseudo-nitzschia* species, some produce the neurotoxin domoic acid (DA), a source of serious health problems for marine organisms. Filter-feeding organisms—e.g., bivalves feeding on toxigenic *Pseudo-nitzschia* spp.—are the main vector of DA in humans. However, little is known about the interactions between bivalves and *Pseudo-nitzschia*. In this study, we examined the interactions between two juvenile bivalve species—oyster (*Crassostrea gigas*) and scallop (*Pecten maximus*)—and two toxic *Pseudo-nitzschia* species—*P. australis* and *P. fraudulenta.* We characterized the influence of (1) diet composition and the *Pseudo-nitzschia* DA content on the feeding rates of oysters and scallops, and (2) the presence of bivalves on *Pseudo-nitzschia* toxin production. Both bivalve species fed on *P. australis* and *P. fraudulenta*. However, they preferentially filtered the non-toxic *Isochrysis galbana* compared to *Pseudo-nitzschia*. The presence of the most toxic *P. australis* species resulted in a decreased clearance rate in *C. gigas*. The two bivalve species accumulated DA in their tissues (up to 0.35 × 10^−3^ and 5.1 × 10^−3^ µg g^−1^ for *C. gigas* and *P. maximus*, respectively). Most importantly, the presence of bivalves induced an increase in the cellular DA contents of both *Pseudo-nitzschia* species (up to 58-fold in *P. fraudulenta* in the presence of *C. gigas*). This is the first evidence of DA production by *Pseudo-nitzschia* species stimulated in the presence of filter-feeding bivalves. The results of this study highlight complex interactions that can influence toxin production by *Pseudo-nitzschia* and accumulation in bivalves. These results will help to better understand the biotic factors that drive DA production by *Pseudo-nitzschia* and bivalve contamination during *Pseudo-nitzschia* blooms.

## 1. Introduction

The pennate diatoms *Pseudo-nitzschia* are cosmopolitan [1,2,3]. About 60 species are currently described, and some of them are considered toxic, i.e., able to produce a neurotoxin—domoic acid (DA) [4,5]. DA is transferred to various organisms within marine food webs when toxic *Pseudo-nitzschia* species are ingested by bivalves (mussels, oysters, scallops), zooplankton (copepods), or planktivorous fish [1,3]. These marine organisms then serve as vectors of the toxin to higher levels of the food web. Bivalves are the primary vector of DA that can cause severe intoxication symptoms in humans [6,7]. Moreover, not only do *Pseudo-nitzschia* species produce DA, but they also excrete it in their environment, e.g., [8,9,10]. Exposure to dissolved DA (dDA) can also have negative effects on the development of marine life [11,12,13].

Despite the harmful effects of DA on higher trophic levels, only few studies have explored the interactions between toxic *Pseudo-nitzschia* and primary consumers. The main interactions studied so far are with copepods, e.g., [14,15,16,17,18,19,20,21,22]. These studies showed no difference in the grazing rates of copepods exposed to toxic and/or non-toxic *Pseudo-nitzschia*. For example, the ingestion rates of *Calanus* copepodites did not differ when exposed to a toxic *P. seriata* or non-toxic *P. obtusa* [20]. In addition, to our knowledge, there are only very few studies available on the influence of toxic *Pseudo-nitzschia* species on bivalve feeding behavior [23,24,25,26]. The results of these studies show that the oyster *Crassostrea virginica* and the mussel *Mytilus edulis* can filter both *P. delicatissima* and *P. multiseries*. However, numerous laboratory studies show that bivalve grazing can be affected by other harmful algal species [27,28,29,30,31,32], but other studies did not evidence any influence on grazing, e.g., [33]. The responses of bivalves to toxin-producing algae are highly species-specific [34].

The influence of primary consumers on *Pseudo-nitzschia* toxin production has mainly been studied with copepods, brine shrimps (*Artemia salina*) or euryhaline rotifers (*Brachionus plicatilis*). However, the influence of filtering bivalves on DA production by *Pseudo-nitzschia* is still poorly known. Copepods can stimulate DA production in some *Pseudo-nitzschia* species [18,20,21], as observed for other algal toxins in harmful dinoflagellate species [35,36,37]. For example, the cellular DA content of *P. seriata* increased from undetected to 13.1 pg cell^−1^ when exposed to the copepods *Calanus hyperboreus* for 8 days [18]. Moreover, the presence of brine shrimps and/or euryhaline rotifers also increases *Pseudo-nitzschia* cellular DA contents, as reviewed in [1]. For example, *A. salina* increased DA production by *P. multiseries* up to 23-fold [38]. Thus, DA production in *Pseudo-nitzschia* can be stimulated by the presence of some primary consumers, but very little is known on the influence of filter feeders on DA production.

*Pseudo-nitzschia* is a common member of the diatom community of the French coasts [39]. Several toxic species have been identified on these coasts [8,40,41,42,43,44] among which are *P. australis*—one of the most toxic species—and *P. fraudulenta*—a less toxic one [45,46]. Along the French coasts, *Pseudo-nitzschia* species grow throughout the year with maximum abundance from May to July and in September [8,39,41,42,44]. However, the interannual variability of *Pseudo-nitzschia* toxic blooms in these ecosystems is related to variations in *Pseudo-nitzschia* species diversity [44]. In France (Bay of Seine, English Channel), DA concentrations in king scallops (*Pecten maximus*) exceeded the European Union regulatory limit (i.e., DA > 20 µg g^−1^ wet weight) for the first time in 2004. Since then, DA contamination events have mainly been reported in scallops, resulting in closures of scallop fisheries on the French coasts [39]. DA may also contaminate other bivalve species [47,48]. Despite the presence of toxic *Pseudo-nitzschia* species in French coastal waters and the health and economic consequences of shellfish DA contaminations, studies on the interactions between *Pseudo-nitzschia* and bivalves are rare.

Considering the absence of information on the influence of bivalves on DA production by *Pseudo-nitzschia* and the importance to better characterize DA accumulation in bivalves for human health issues, the aims of this study were to investigate the interactions between two juvenile bivalve species—the oyster *C. gigas* and the scallop *P. maximus*—and two *Pseudo-nitzschia* species—*P. australis* and *P. fraudulenta*—from French coastal waters. More specifically, we investigated (1) if bivalve clearance and filtration rates varied according to the algal species and the *Pseudo-nitzschia* toxin content, (2) to which extent bivalves accumulated DA depending on the *Pseudo-nitzschia* species they filtered, and (3) if the presence of bivalves influenced DA production by the two *Pseudo-nitzschia* species with contrasting toxin contents.

## 2. Results

### 2.1. Cell Concentrations, Clearance Rates, Filtration Rates

#### 2.1.1. *Crassostrea gigas*

Condition 1: Juvenile oysters exposed to *P. fraudulenta* and *I. galbana* for 5 days. *I. galbana* and *P. fraudulenta* cell concentrations decreased over time in all replicates in the presence of *C. gigas* (Figure 1A), whereas they increased in the absence of oysters (data not shown). The cell concentrations decreased to exhaustion between day 1 and day 2 for *I. galbana* and between day 3 and day 4 for *P. fraudulenta* (Figure 1A). The clearance rate (CR) of juvenile *C. gigas* was around 0.43 mL h^−1^ ind^−1^ on days 1 and 2, then it gradually increased significantly from 0.71 mL h^−1^ ind^−1^ on day 3 to 2.92 mL h^−1^ ind^−1^ on day 5 (RM ANOVA, *p* < 0.0001, Figure 1B). It started to increase when the cell concentration of the microalgae was less than 3.5 × 10^3^ cells mL^−1^ (Figure 1A,B). For *I. galbana*, the filtration rate (FR) decreased significantly from 16 × 10^3^ cells h^−1^ ind^−1^ (day 1) to 3.7 × 10^3^ cells h^−1^ ind^−1^ (day 2) (RM ANOVA, *p* < 0.01, Figure 1C). For *P. fraudulenta*, the FR decreased significantly from 9.5 × 10^3^ cells h^−1^ ind^−1^ on day 1 to 2.5 × 10^3^ cells h^−1^ ind^−1^ on day 4 (RM ANOVA, *p* < 0.01, Figure 1D). This decreased FR was related to the decreasing cell concentrations of both algae throughout the experiment. In addition, the maximum FR for *I. galbana* was significantly higher than the maximum FR for *P. fraudulenta* (ANOVA, *p* < 0.05).

Condition 2: Juvenile oysters exposed to *P. australis* and *I. galbana* for 5 days. In the presence of oyster spat, *I. galbana* cell concentrations decreased to depletion between day 1 and day 2 (Figure 1E). In contrast, *P. australis* cell concentrations started to decrease only after day 1 and until day 3, and then it stabilized around 8 × 10^3^ cells mL^−1^ until the end of the experiment (Figure 1E). The cell concentrations of all two microalgae increased in the oyster-free control (data not shown). The CR remained quite constant and low throughout the 5 days of the experiment, between 0.14 ± 0.43 and 0.40 ± 0.10 mL h^−1^ ind^−1^ (Figure 1F). Juvenile *C. gigas* exposed to *P. australis* did not increase their CR as microalgal concentrations decreased (Figure 1F). The FRs for *I. galbana* and *P. australis* decreased significantly from 30 × 10^3^ (day 1) to 3 × 10^3^ cells h^−1^ ind^−1^ (day 2) and from 8.7 × 10^3^ cells h^−1^ ind^−1^ on day 2 to 0.9 × 10^3^ cells h^−1^ ind^−1^ on day 4, respectively (RM ANOVA, *p* < 0.01, Figure 1G,H). On day 1, the FR for *P. australis* was zero (Figure 1H).

Conditions 3 and 4: Juvenile oysters exposed to *P. fraudulenta* (condition 3) or *P. australis* (condition 4) for 5 days. *P. fraudulenta* and *P. australis* cell concentrations decreased over time in the presence of *C. gigas* (Figure 2A), whereas they increased in the absence of oysters (data not shown). In the presence of *C. gigas*, the cell concentration of *P. australis* decreased between day 0 and day 3, and then it stabilized until the end of the experiment around 8.5 × 10^3^ cells mL^−1^, while there were no *P. fraudulenta* cells left at the end of the experiment (Figure 2A). The CR of *C. gigas* exposed to *P. fraudulenta* was between 0.30 and 0.55 mL h^−1^ ind^−1^ from day 1 to day 4, and then increased significantly up to 3.55 mL h^−1^ ind^−1^ on day 5 (RM ANOVA, *p* < 0.001, Figure 2B). The FR for *P. fraudulenta* remained quite constant between day 1 and day 3 with 10.8 × 10^3^ ± 5.5 × 10^3^ (day 1) and 6.2 × 10^3^ ± 1.9 × 10^3^ cells h^−1^ ind^−1^ (day 3, Figure 2B). Then, it decreased significantly to 1.1 × 10^3^ cells h^−1^ ind^−1^ (day 4) and stabilized at 1.7 × 10^3^ cells h^−1^ ind^−1^ on day 5 (RM ANOVA, *p* < 0.001, Figure 2B). The CR of *C. gigas* exposed to *P. australis* varied between 0 ± 0.13 and 0.43 ± 0.13 mL h^−1^ ind^−1^ (Figure 2C) and did not differ significantly throughout the experiment. The FR for *P. australis* was relatively constant during the first three days with an average of 6.2 × 10^3^ ± 0.4 × 10^3^ cells h^−1^ ind^−1^, and then decreased significantly down to 1.2 × 10^3^ cells h^−1^ ind^−1^ on day 4 and 0 cells h^−1^ ind^−1^ on day 5 (RM ANOVA, *p* < 0.01, Figure 2C), while *P. australis* cells were still present. The maximum CR was significantly lower for *P. australis* than for *P. fraudulenta* (ANOVA, *p* < 0.001; Figure 2B,C).

#### 2.1.2. *Pecten maximus*

Condition 1: Juvenile scallops exposed to *P. fraudulenta* and *I. galbana* for 6 h. *I. galbana* cell concentrations decreased over time from 45.2 × 10^3^ cells mL^−1^ after 1 h to 7.2 × 10^3^ cells mL^−1^ after 6 h (Figure 3A). *P. fraudulenta* cell concentrations remained constant during the first 2 h with an average 43.7 × 10^3^ cells mL^−1^, and then the concentrations decreased until the end of the experiment when they reached 3.3 × 10^3^ cells mL^−1^ (Figure 3A). The CR of juvenile *P. maximus* was relatively constant during the first 4 h, between 20 ± 15 and 70 ± 35 mL h^−1^ ind^−1^. Then, it increased significantly to 180 mL h^−1^ ind^−1^ at T = 5 h and 130 mL h^−1^ ind^−1^ at T = 6 h (RM ANOVA, *p* < 0.01, Figure 3B). This increased CR coincided with cell concentrations of less than 18.5 × 10^3^ cells mL^−1^ in the culture medium (Figure 3A,B). The FR for *I. galbana* and *P. fraudulenta* did not show any significant difference throughout the 6 h of *P. maximus* exposure because variability among replicates was high. The FRs were between 900 × 10^3^ ± 730 × 10^3^ and 2140 × 10^3^ ± 1050 × 10^3^ cells h^−1^ ind^−1^ for *I. galbana* (Figure 3C) and between 0 ± 560 × 10^3^ and 3600 × 10^3^ ± 1300 × 10^3^ cells h^−1^ ind^−1^ for *P. fraudulenta* (Figure 3D).

Condition 2: Juvenile scallops exposed to *P. australis* and *I. galbana* for 6 h. *I. galbana* cell concentrations decreased over time from 43.8 × 10^3^ cells mL^−1^ at the beginning of the experiment to 7.6 × 10^3^ cells mL^−1^ after 6 h (Figure 3E). *P. australis* cell concentrations remained constant during the first 3 h of exposure with an average 44.7 × 10^3^ cells mL^−1^, and then decreased to 3.8 × 10^3^ cells mL^−1^ after 6 h of exposure (Figure 3E). The CR of *P. maximus* was relatively constant during the first 4 h, between 16 ± 20 and 82 ± 24 mL h^−1^ ind^−1^, and then increased significantly to reach 230 mL h^−1^ ind^−1^ at T = 5 h (RM ANOVA, *p* < 0.05) and decreased significantly to 75 mL h^−1^ ind^−1^ at T = 6 h (RM ANOVA, *p* < 0.05, Figure 3F). The FR for *I. galbana* and *P. australis* did not show any significant difference throughout the 6 h of *P. maximus* exposure because of high variability among replicates. The FRs were between 290 × 10^3^ ± 190 × 10^3^ and 2300 × 10^3^ ± 780 × 10^3^ cells h^−1^ ind^−1^ for *I. galbana* (Figure 3G) and between 30 × 10^3^ ± 2210 × 10^3^ and 5280 × 10^3^ ± 1760 × 10^3^ cells h^−1^ ind^−1^ for *P. australis* (Figure 3H).

### 2.2. DA Accumulation in *Crassostrea gigas* and *Pecten maximus*

At the beginning of the experiment, the DA levels in the flesh tissues of all bivalves were below the detection limit of the ELISA method (data not shown). After 5 days of *C. gigas* exposure (Figure 4A) and 6 h of *P. maximus* exposure to *Pseudo-nitzschia* cells (Figure 4B), significant DA concentrations were measured in all the flesh tissues of the bivalves.

No significant differences were observed between the DA levels in *C. gigas* exposed to single *Pseudo-nitzschia* cultures and *C. gigas* exposed to mixed cultures of the two *Pseudo-nitzschia* species. In contrast, significant differences in DA accumulation were observed in *C. gigas* exposed to *P. fraudulenta* and *P. australis*. *C. gigas* exhibited significantly more DA when exposed to *P. australis* (0.32 × 10^−3^ to 0.35 × 10^−3^ µg DA g^−1^) than when exposed to *P. fraudulenta* (0.18 × 10^−3^ to 0.21 × 10^−3^ µg DA g^−1^; ANOVA, *p* < 0.05, Figure 4A). *P. maximus* DA levels were also significantly higher following exposure to *P. australis* than following exposure to *P. fraudulenta*, with 5.1 × 10^−3^ and 1.2 × 10^−3^ µg DA g^−1^ on average, respectively (ANOVA, *p* < 0.05, Figure 4B). No significant relationship was found between the weight of each bivalve species and its DA content (data not shown).

### 2.3. DA Concentrations in *Pseudo-nitzschia* Cells (cDA) and in the Medium (dDA), and Nutrient Concentrations

In the experiments with *C. gigas*, the initial cDA concentration in *P. fraudulenta* was 1.6 fg cell^−1^ (data not shown). At the end of the experiment (i.e., at T = 96 h), it was 0.9 fg cell^−1^ in the control, 1.9 fg cell^−1^ in *P. fraudulenta* in mixed culture in the presence of *C. gigas*, and 53.3 fg cell^−1^ in *P. fraudulenta* in single culture in the presence of *C. gigas* (Figure 5A), with average *P. fraudulenta* cell concentrations of 1.5 × 10^5^ cells mL^−1^, 5 × 10^2^ cells mL^−1^, and 1 × 10^3^ cells mL^−1^, respectively. *P. fraudulenta* cDA was more than 58-fold higher than the control level when it was in single culture in the presence of *C. gigas*, and more than 28-fold higher than when it was in mixed culture (ANOVA, *p* < 0.001, Figure 5A). As far as *P. australis* is concerned, the initial cDA concentration was 4.2 fg cell^−1^ (data not shown). At the end of the experiment (i.e., at T = 120 h), the cDA concentration was 5.6 fg cell^−1^ in the control, 28.3 fg cell^−1^ when *P. australis* was in mixed culture in the presence of *C. gigas*, and 79.9 fg cell^−1^ when it was in single culture in the presence of *C. gigas* (Figure 5B), with average *P. australis* cell concentrations of 5.9 × 10^4^, 8 × 10^3^, and 8.5 × 10^3^ cells mL^−1^, respectively. The cDA concentration in *P. australis* was significantly different, i.e., 5-fold or 14-fold higher, when *P. australis* was in mixed culture (ANOVA, *p* < 0.05) or in single culture to *C. gigas* compared to the control, respectively (ANOVA, *p* < 0.001, Figure 5B).

In the presence of *P. maximus*, the initial cDA concentration in *P. fraudulenta* was 1.6 fg cell^−1^ (data not shown). After 6 h, it was 1.7 fg cell^−1^ in the control and 41.3 fg cell^−1^ when *P. maximus* was exposed to *P. fraudulenta* in mixed culture (Figure 5C) with a final *P. fraudulenta* cell concentration of 3.3 × 10^3^ cells mL^−1^. *P. fraudulenta* cDA was 24-fold higher than in the control condition when it was in the presence of *P. maximus* (ANOVA, *p* < 0.001, Figure 5C). The initial *P. australis* cDA concentration was 9.7 fg cell^−1^ (data not shown). After 6 h, it was 22.2 fg cell^−1^ in the control and 676.1 fg cell^−1^ in *P. australis* in mixed culture in the presence of *P. maximus* (Figure 5D), with a final *P. australis* cell concentration of 7.6 × 10^3^ cells mL^−1^. *P. australis* cDA was significantly different and more than 30-fold higher than in the control after 6 h in the presence of *P. maximus* (ANOVA, *p* < 0.001, Figure 5D).

When *C. gigas* and *P. maximus* were exposed to *P. fraudulenta*, dDA concentrations at the beginning and at the end of the experiment were not significantly different in the control and in all the conditions tested (Appendix A). During exposure to *P. australis*, the dDA concentrations in all treatments with or without *C. gigas* were not significantly different from each other either (Appendix A). In contrast, when *P. maximus* was exposed to *P. australis*, the dDA concentrations were significantly higher at the end of the experiment compared to the beginning or to the control, with 4.13, 2.15, and 2.33 pg mL^−1^, respectively (ANOVA, *p* < 0.05, Appendix A).

The nutrient measurements revealed relatively stable nitrate and phosphate concentrations throughout all experiments (Appendix A). In contrast, the silicate concentrations significantly decreased in all but one condition between the beginning and the end of the experiments (ANOVA, *p* < 0.05, Appendix A): when *C. gigas* was exposed to *P. australis* (condition 4), the silicate concentration was significantly lower in the control compared to the silicate concentration measured at the end of the experiment (Appendix A). However, the final silicate concentrations under the other conditions were not significantly different from those in the control condition. In addition, nutrients (nitrate, phosphate, or silicate) were constantly available at high concentrations in the culture medium and they were never depleted at the end of the experiments. At that time and in all conditions, the minimum concentrations of nitrate, phosphate, and silicate were 117, 7, and 32 µmol L^−1^, respectively (Appendix A).

### 2.4. Potential Relationship between cDA Concentrations in *P. australis* and *P. fraudulenta* on the One Hand, and the CR, the FR, and DA Accumulation in Bivalve Species on the Other Hand

Figure 6 and Appendix A present the relationships between the cDA concentrations in *P. australis* (Figure 6) and *P. fraudulenta* (Appendix A) and the average CR and FR of *C. gigas* throughout the 5 days of the experiment. The results show no relationship between *P. fraudulenta* cDA concentrations and the CR or FR of *C. gigas* (Appendix A). In contrast, *P. australis* cDA concentrations present linear relationships with the CR (r^2^ = 0.98; CR = −709 cDA + 235; Figure 6A) and the FR (r^2^ = 0.92; FR = 0.0202 cDA − 33; Figure 6B) of *C. gigas*.

As far as *C. gigas* DA concentrations at the end of the experiment are concerned, no relationship was found with their FR of any of the *Pseudo-nitzschia* species (data not shown). In contrast, Figure 7 shows a linear regression between *C. gigas* DA concentrations and *P. australis* cDA concentrations at the end of the experiment (r^2^ = 0.47; *C. gigas* cDA concentration = 1.06 *P. australis* cDA concentration + 277.11). In contrast, no significant relationship was found for *P. fraudulenta*. As for *P. maximus,* only one experiment was performed, so that data are not sufficient to allow us to explore any relationship.

## 3. Discussion

### 3.1. Feeding Responses of the Bivalves Exposed to *Pseudo-nitzschia* Species

The feeding behavior of bivalves is generally dependent on the phytoplankton species available [27,49]. Our results confirm this observation: the two studied bivalves filtered the haptophyte *I. galbana* rather than the two *Pseudo-nitzschia* species. These results confirm the preference observed in juvenile *Crassostrea virginica* oysters by [24] and [26]. These studies report that *C. virginica* filtered other microalgal species (*Ditylum brightwellii*, *Thalassiosira weissflogii*, *I. galbana*) rather than *Pseudo-nitzschia* (*P. delicatissima* and *P. multiseries*). Nevertheless, when *C. virginica* was exposed to toxic *P. multiseries* and non-toxic *P. delicatissima*, their clearance rates were lower than when they were exposed to other diatoms, but similar for both *Pseudo-nitzschia* species regardless of their toxin level [26]. Our observation that *C. gigas* rather fed on the smaller *I. galbana* cells than on the large, needle-shaped *Pseudo-nitzschia*, regardless of their toxin content, is also consistent with more general observations indicating that filtration by bivalves is influenced by the physical aspects of the microalgae, i.e., size, density, electric charge, and morphology [49,50]. The preferential choice of some microalgae suggests that bivalves possess mechanisms in their gills and/or labial palps that select algal species based on cell size and/or morphology [25,49]. However, the selection of microalgae may also depend on a chemosensory response to food stimuli on the gills and labial palps of bivalves [49,51]. The mechanism potentially involved in the chemosensory detection of toxic cells by bivalves remains to be determined [31].

The present study also demonstrates changes in the clearance and filtration rates of bivalves fed on distinct *Pseudo-nitzschia* species. Decreased clearance rates of bivalves (oysters, mussels, clams, scallops) have been reported following exposure to the toxic dinoflagellates *Alexandrium* spp. and *Karenia brevis* [29,32,33,52,53]. However, clams (*Mulinia edulis*) and mussels (*Mytilus chilensis*) filtered *Alexandrium catenella*—a producer of paralytic shellfish toxin (PST)—and *Alexandrium affine*—a non-producer of PST—at similar rates [33]. We observed lower clearance rates when *C. gigas* was exposed to the most toxic species *P. australis* compared to the less toxic *P. fraudulenta*. As both *Pseudo-nitzschia* species had similar morphologies and cell sizes, the lower clearance rate for *P. australis* is attributable to its higher cellular DA content. Therefore, DA affected the feeding behavior of *C. gigas*, as already shown for paralytic shellfish toxins produced by the dinoflagellate *Alexandrium tamarense* [27]. These results confirm that the feeding response to diverse microalgae is species-specific in bivalves, since DA did not affect another oyster species (*C. virginica*) that filters toxic and non-toxic *Pseudo-nitzschia* at similar rates [24,25,26]. Moreover, oysters are more sensitive than other bivalves to phycotoxin-producing microalgae [54]. This is supported by our results: the presence of either *Pseudo-nitzschia* species did not affect the clearance or filtration rates of *P. maximus*. Consequently, DA did not affect the feeding behavior of scallop, as opposed to recent results about *Alexandrium minutum* and PST [55]. The presence of *Pseudo-nitzschia* cells affected the feeding behavior of *C. gigas* more than that of *P. maximus*. However, our experiments with *P. maximus* were shorter than those with *C. gigas,* and the response of *P. maximus* over more than a few hours will have to be investigated to confirm our results. Furthermore, to really grasp the real impact of *Pseudo-nitzschia* on the feeding dynamics of *C. gigas* and *P. maximus*, further studies are needed to precisely explore the pre-ingestive feeding processes (e.g., rejection of cells in the pseudofeces). These processes may also influence the ingestion of *Pseudo-nitzschia* cells beyond filtration, as shown for the oyster *C. virginica* by [25].

### 3.2. DA Accumulation in the Bivalves

*C. gigas* and *P. maximus* both retained DA when fed on toxic *P. australis* or *P. fraudulenta* cells. The maximum DA concentrations accumulated by *C. gigas* and *P. maximus* were measured when they were exposed to *P. australis*. The duration of exposure, the experimental conditions, the bivalve species, and bivalve size vary across studies, so that it is difficult to compare our DA accumulation results with literature data. However, the maximum DA concentrations accumulated by oysters (0.35 × 10^−3^ µg g^−1^ after 5 days of exposure) and scallops (5.1 × 10^−3^ µg g^−1^ after 6 h of exposure) in the present study are lower than the maximum DA levels measured in previous studies with various bivalves during exposure to *Pseudo-nitzschia* under controlled conditions: a maximum of 22.8 µg g^−1^ was recorded for the oyster *C. virginica* after 14 days of exposure to *P. multiseries* [23], and 3.1 × 10^3^ µg g^−1^ in the digestive gland of the Atlantic sea scallop *Placopecten magellanicus* after 22 days of exposure to *P. multiseries* [56]. The lower toxin accumulation measured in our study probably resulted from the shorter exposure times, the single supply and non-renewal of toxic *Pseudo-nitzschia*, and the lower DA concentration in *Pseudo-nitzschia*. The *P. multiseries* strains used in [56] and [23] produced 1000 times more DA than *P. australis* and *P. fraudulenta* did in this study (up to 6.67 pg cell^−1^ [56]; up to 9.8 pg cell^−1^ [23]). The two bivalve species were also exposed to a continuous supply of *P. multiseries,* whereas no *P. australis* or *P. fraudulenta* were added after the beginning of the experiments in our study, hence a very low number of toxic cells per bivalve. Furthermore, toxic cell ingestion by oysters mostly occurred in the first three days of exposure, but we measured DA contents after five days; therefore, toxin depuration may have been ongoing within oyster tissues when the measurements were made. The authors of [57] indeed highlight that *C. gigas* accumulate DA within a few hours and start depurating it immediately: oysters contaminated up to 36.3 µg g^−1^ of DA had fully depurated it after 120 h. Our oysters may have been more contaminated at the beginning of our experiment than after 5 days, when samples were taken for DA measurements. Finally, the present study was performed with oyster spat and juvenile scallops, whereas most of the literature data relate to adult bivalves.

The different levels of DA accumulation by the two bivalves may partly result from differences in clearance and filtration rates. *P. maximus* filtered more toxic *Pseudo-nitzschia* cells than *C. gigas* did: 16 × 10^5^ cells h^−1^ ind^−1^ or 16 × 10^5^ cells h^−1^ g^−1^ versus 3.7 × 10^3^ cells h^−1^ ind^−1^ or 2 × 10^5^ cells h^−1^ g^−1^ on average, respectively. The authors of [23] obtained similar results with juvenile oysters (*C. virginica*), which accumulated 3–75 times less DA than juvenile mussels (*Mytilus edulis*) did, while the CRs of oysters were 7.4–8.5 times lower than those of mussels. The different DA accumulation rates of the two bivalves could also be linked to differences in gill anatomy and in the sorting system of digestive particles. The size of the gill filaments may be a physical limitation to the selection of micro-algal particles based on their cell size [58]. Scallops have larger branchial filaments (200 µm [59]) compared to oysters (<70 µm [25]) allowing easier ingestion of longer cells like *Pseudo-nitzschia*. The selective rejection of *Pseudo-nitzschia* based on size may be another explanation for the differences in DA accumulation in our study given the mean size of *P. australis* cells (52 µm). This hypothesis stresses once more the need to complete the present study with further observations, e.g., pseudofeces formation, as in [25,26], to determine whether the differences in DA concentrations in the bivalves are linked to differences in *Pseudo-nitzschia* ingestion. In addition, the differential DA accumulation by oysters and scallops may also be explained by differential DA metabolism and degradation by intestinal bacteria during toxin accumulation [60]. Therefore, to better comprehend the differences in DA accumulation, it would be interesting to investigate the ways *C. gigas* and *P. maximus* ingest *Pseudo-nitzschia* and degrade DA. DA accumulation by *C. gigas* exposed to *P. australis* was also correlated to cDA concentrations in *P. australis*. To our knowledge, this is the first time that a relationship between toxin accumulation by bivalves and the toxin content per microalgal cell is highlighted. Thus, beyond all the processes likely to occur in *C. gigas* exposed to *P. australis* (altered feeding behavior, possible rejection in the pseudofeces, metabolization of DA in the digestive tract), DA accumulation seems greatly influenced by the toxin content of the algal cells. This suggests that the toxin content of *Pseudo-nitzschia* cells is critical for determining the toxin content of *C. gigas* in situ, besides the extent of the *Pseudo-nitzschia* bloom itself.

### 3.3. Induction of DA Production in *Pseudo-nitzschia* in the Presence of Bivalves

During the different co-exposure experiments, the cellular DA content of *P. australis* and *P. fraudulenta* in the exponential growth phase without any nutrient limitation increased 14- and 58-fold, respectively, in the presence of *C. gigas,* and 30- and 24-fold in the presence of *P. maximus* compared to the bivalve-free control. In the literature, the increase in cDA is mainly related to changes in environmental factors, especially nutrient availability (reviewed in [1,2]). Most *Pseudo-nitzschia* species significantly increase their cDA content in the stationary phase under silicate or phosphate limitation [45,61,62,63,64,65,66]. In the present study, given that nitrate, phosphate, and silicate were replete at the end of our experiments, it is unlikely that changes in cDA in *P. australis* and *P. fraudulenta* were related to nutrient concentrations. The experiments were performed in a controlled environment, so that no other parameter (e.g., temperature or light intensity) varied during exposure. Therefore, the observed increase in cDA was related to the presence of the bivalves. This suggests that filter-feeding bivalves can stimulate toxin production by *P. australis* and *P. fraudulenta*.

To our knowledge, these are the first results showing that DA production by *Pseudo-nitzschia* can increase in the presence of filter-feeding bivalves. The presence of bivalves induced an increase in *Pseudo-nitzschia* cDA up to 58-fold in *C. gigas* within 5 days and up to 30-fold in *P. maximus* within only 6 h. Similar results for PST production in the dinoflagellate *Alexandrium fundyense* were obtained in the presence of *Mytilus edulis* and *Mya arenaria* [67]. Increased toxin production by *Pseudo-nitzschia* in the presence of primary consumers other than bivalves has been reported in the literature: DA production increased or was induced in some *Pseudo-nitzschia* species in the presence of herbivorous copepods [17,18,20,21,68]. Furthermore, only 2 h of exposure to copepods resulted in a cDA increase in *P. seriata* [18,20]. Our results show that *Pseudo-nitzschia* can increase their toxin production not only in the presence of copepods, but also when exposed to other primary consumers like bivalves. We also confirm that exposure times in the range of a few hours are sufficient to influence DA production. Complex interactions have already been shown between primary consumers and toxin-producing phytoplankton, mainly between copepods and *Pseudo-nitzschia* or between bivalves and dinoflagellates. Some dinoflagellates can increase their toxin production in the presence of copepods and bivalves, and diatoms react to the presence of copepods through different morphological and biochemical defense strategies. However, the relationship between toxin production and grazing or filtration by bivalves is still poorly understood [69,70], and not fully explored for *Pseudo-nitzschia*. The relationships observed in this study between the clearance rates of oysters and *Pseudo-nitzschia* cDA contents, and also between filtration rates and cDA contents, suggest complex interactions probably mediated by chemical communication. The feeding-related cues may be metabolites released by the bivalves upon filtration or ingestion of *Pseudo-nitzschia* cells. DA may also act as a chemical cue, although dDA concentrations did not significantly vary during our experiments. Metabolites excreted by grazers can warn harmful algae of their presence [67]. For example, primary consumers such as copepods excrete copepodamides, a group of polar lipids that induce and/or stimulate the production of paralytic shellfish toxins by *Alexandrium* and DA production by *Pseudo-nitzschia* [69,71]. The reciprocal influence between *P. fraudulenta*/*P. australis* and *C. gigas*/*P. maximus* observed in the present study shows that further metabolomics studies are needed to explore the mechanisms and the chemical cues associated with the stimulation of DA production and the alteration of the feeding behavior during bivalve/*Pseudo-nitzschia* interactions.

## 4. Conclusions

This study characterizes the interactions between filter-feeding bivalves and two *Pseudo-nitzschia* species. *C. gigas* and *P. maximus* were able to feed on *P. australis* and *P. fraudulenta* and accumulate DA in their tissue, even during short-term exposure. Furthermore, the presence of bivalves induced an increase in cDA content in both *Pseudo-nitzschia* species, suggesting that DA production by *Pseudo-nitzschia* could be a grazer-deterrent mechanism. This is also supported by the facts that (1) both bivalves preferentially filtered the non-toxic *I. galbana* and (2) the presence of the most toxic *P. australis* affected the clearance rate of *C. gigas*. However, the influence of *Pseudo-nitzschia* on the feeding behavior was bivalve-specific: *C. gigas* was more affected than *P. maximus* was. These results will help to better understand the biotic factors that control DA production by *Pseudo-nitzschia* and bivalve contamination during *Pseudo-nitzschia* blooms. These findings are a first step before further studies on the feeding behavior of bivalves and their metabolism when exposed to toxic *Pseudo-nitzschia*, and also on the chemical communication underlying the interactions between bivalves and toxic diatoms.

## 5. Materials and Methods

### 5.1. Organisms

#### 5.1.1. Phytoplankton Cultures

Three species of microalgae were tested: two species belonging to the genus *Pseudo-nitzschia* and one species belonging to Prymnesiophyceae, *Isochrysis galbana*. By using the two *Pseudo-nitzschia* species, we tested two algae with similar shapes and sizes but different DA contents. By adding *I. galbana*, we also compare the interactions of the bivalves the two DA-producing species on the one hand and one non-producer on the other hand.

*Pseudo-nitzschia* cultures. Two species of *Pseudo-nitzschia* were used: *P. australis* (P6B3) and *P. fraudulenta* (PNfra 12). *P. australis* strain P6B3 is a known toxic strain [45,46], while PNfra 12 is less toxic (unpublished data). Strain P6B3 was isolated from the West Finistère coast of France (Brittany, Atlantic coast) in April 2014, and strain PNfra 12 was isolated from the Bay of Seine (Normandy, English Channel) in August 2011. To establish monoclonal strains, single cells were isolated using a micropipette, washed three times with filter-sterilized seawater, and incubated in 4-well culture plates in K/2-medium [72] enriched in Si(OH)_4_ (54 µmoL L^−1^) at a temperature of 16 °C, an irradiance of 30 µmoL photons m^−2^ s^−1^, and a 14 h:10 h light:dark (L:D) cycle. When the clonal culture was established, it was maintained in 50 mL ventilated plastic flasks (Falcon^®^, Corning Life Sciences, Tewksbury, MA USA) in K/2-medium + Si(OH)_4_ in the same conditions. Strain P6B3 was identified using sequencing of the gene ITS1-5.8S-ITS2. This strain had identical sequences to *P. australis* strains from [45]. Strain PNfra 12 was identified from measurements of frustule properties by transmission electron microscopy (TEM). For TEM observations, culture samples were cleaned up to remove organic material according to the method of [73], except that 2 mL of hydrochloric acid were used in addition to sulfuric acid [44]. Drops of cleaned material were placed on grids and studied in a JEOL-1010 (Jeol, Tokyo, Japan) electron microscope operating at 100 kV. For morphometric measurements, the length and width of the valves were measured, together with the densities of their striae, fibulae, and poroids, the number of rows of poroids, and the absence/presence of a central interspace. A minimum of 10 cells was measured for each strain. For cell length, each strain was observed before each experiment under a Nikon Eclipse 80i light microscope equipped with a Nikon DS-Ri2 camera, and 20 cells were measured (length in µm) using NIS-Elements Imaging Software. Cell length was calculated as the mean ± standard deviation of these 20 cells. Cell length was 52 ± 2.5 µm for P6B3, and 47 ± 1 µm for PNfra 12.

*Isochrysis galbana* cultures. The non-toxic flagellate *I. galbana* (AC34) was obtained from the Algobank-Caen culture collection of the University of Caen Normandy. This species is routinely used as a food source for bivalves. AC34 was cultured in K/2-medium enriched in Si(OH)_4_ (54 µmoL L^−1^) at a temperature of 16 °C, an irradiance of 30 µmol photons m^−2^ s^−1^, and a 14 h:10 h L:D cycle.

#### 5.1.2. Juvenile Filter-Feeding Bivalves

One-year-old juvenile scallops (*P. maximus*) were obtained from Tinduff hatchery (Finistère, Brittany, France). The average shell length was 4.10 ± 0.75 cm, and the average wet weight 1.0 ± 0.2 g.

Oyster (*C. gigas*) spat at stage T6 was obtained from France Naissain (Vendée, Loire Countries, France). The average shell length was 1.0 ± 0.15 cm, and the average wet weight 18 ± 4 mg.

### 5.2. Experimental Procedure

The aim of our experiments was to study the early effects of exposure of juvenile filtering bivalves to toxic *Pseudo-nitzschia*, without prior acclimatization of the bivalves to *Pseudo-nitzschia*. Algal cultures were not renewed during the experiments in order to study the response of the bivalves along with the changes in the proportions of the different microalgae. Before each experiment, each phytoplankton strain and bivalve species was acclimated to the experimental conditions, i.e., 15 °C, 100 µmol photons m^−2^ s^−1^, and a 14 h:10 h L:D cycle.

#### 5.2.1. Experimental Protocols of Contact between Bivalves and Microalgae

The experiments were carried out in Erlenmeyer flasks with an oxygenation system. Four experimental protocols (Table 1) were used with the bivalve species, the microalgal species, or a combination of species. A bivalve-free control condition was prepared for each experimental condition. The experiments were carried out in triplicates.

For *C. gigas* spat, 20 individuals were added in 250 mL of K/2-medium + Si(OH)_4_ (95 µM) for five days. For juvenile *P. maximus* experiments, scallops were exposed for 6 h: one juvenile was added in 220 mL of K/2-medium + Si(OH)_4_ (63 µM). Considering the FR and that no microalgae culture was added during the experiments, *P. maximus* was not exposed for more than 6 h since no more microalgae were available after this time.

#### 5.2.2. Sampling

At the beginning of the experiments, samples containing all the tissues of 20 *C. gigas* on the one hand and three *P. maximus* on the other hand were taken to determine the DA concentrations in their flesh tissues. Microalgal culture samples were also taken to determine extracellular dissolved DA (dDA), cellular DA (cDA) and nutrient concentrations. Samples of the medium containing the microalgae were collected every day (from day 0 to day 5) for *C. gigas* and every hour (from hour 0 to hour 6) for *P. maximus* to determine microalgal cell concentrations and calculate the clearance and filtration rates. At the end of the experiments, the bivalves of each flask were collected to determine DA concentrations in their flesh tissues, and the medium was also sampled for nutrient concentration measurements. In the flasks containing the *Pseudo-nitzschia* species, the medium was sampled to determine dDA and cDA concentrations. The samples for the dDA and cDA measurements when *C. gigas* was exposed to *P. fraudulenta* were taken on day 4 instead of day 5 to avoid having no *P. fraudulenta* cells left in the culture medium at the end of the experiment.

### 5.3. Data Analysis

#### 5.3.1. Cell Concentrations, Clearance Rates, Filtration Rates

*P. australis* and *P. fraudulenta* cell concentrations were estimated using a Nageotte counting chamber, and *I. galbana* cell concentrations were estimated using a Malassez counting chamber. Then, we determined the clearance rates of the bivalves (CR in mL h^−1^ ind^−1^)—every day for 5 days and every hour for 6 h for the experiments with *C. gigas* and *P. maximus*, respectively—using the following equation [74] and assuming the absence of sedimentation and negligible growth in the Erlenmeyer flasks:(1)$$\mathrm{CR}=\ln(\frac{C_{1}}{C_{2}})\;\times \;\frac{V}{T_{2}\;-\;T_{1}}\;\times \;\frac{1}{N}$$
where C_1_ and C_2_ are the phytoplankton cell concentrations in the presence of the bivalves (cells mL^−1^) at T_1_ and T_2_ (hours), V is the volume of medium in the Erlenmeyer flasks (mL), and N is the number of bivalve individuals in each Erlenmeyer flask.

The filtration rates (FR, in cells h^−1^ ind^−1^) were determined from the previous equation:(2)$$\mathrm{FR}\;=\;\mathrm{CR}\;\times \;\frac{C_{1}{\;+\;C}_{2}}{2}$$

CR and FR were calculated for each bivalve only over the period when phytoplankton cells were not depleted in the culture medium.

#### 5.3.2. Dissolved Inorganic Nutrient Analysis

Samples for the determination of inorganic nutrient (nitrate, phosphate and silicate) concentrations in the culture medium were obtained by filtering 10 mL of culture medium on a 0.45 µm cellulose acetate membrane under low filtration pressure to remove algal cells. The filtrate was stored at −20 °C for nitrate and phosphate assays and at 4 °C for the silicate assay prior to analysis. Dissolved nutrients were quantified with an auto-analyzer AA3_HR (Seal Analytical) following standard protocols [75].

#### 5.3.3. Domoic Acid Analysis

DA in *Pseudo-nitzschia* species. Total (tDA) and dissolved (dDA) domoic acid concentrations were measured using enzyme-linked immunosorbent assay (ELISA) Biosense kits (Biosense Laboratories, Bergen, Norway), following the manufacturer’s instructions modified from [76]. tDA was measured from 5 mL of culture stored at −20 °C until analysis (maximum two months). For dDA, 5 mL of culture were centrifuged, and the supernatant was frozen at −20 °C prior to analysis (maximum two months). The dDA samples were directly analyzed. For tDA, the whole culture samples were sonicated on ice with a sonication probe (Bioblock Scientific Vibracell 72442 ultrasons) for 4 min to disrupt cell membranes and release DA from the cells. Then, the samples were filtered on a 0.2 µm cellulose acetate membrane to remove cell debris, and the filtrate was analyzed. Every sample was analyzed by ELISA in duplicate for quality control purposes, as specified by the manufacturer. cDA (pg cell^−1^) was obtained by subtracting dDA from tDA and by normalizing to the cell concentration.

DA in bivalves. Three sets of 20 *C. gigas* or 3 *P. maximus* were collected from each experimental protocol for analysis at the beginning and the end of the experiment. These pools of bivalves were frozen at −20 °C prior to analysis. The bivalves were dissected, weighed, placed in a microtube (Eppendorf, Germany) containing the extraction solution (50%/50% methanol/water), and crushed with a pellet piston (Eppendorf, Germany). The samples were vigorously shaken on a vortex for 1 min and centrifuged at 3000 g for 10 min at room temperature to recover the supernatant for DA analysis. DA quantifications were then carried out as presented above, using an ELISA kit (Biosense Laboratories, Bergen, Norway). The DA concentration in bivalve tissues (µg DA g^−1^) was calculated using the following equation:(3)$$\mathrm{DA}\;\mathrm{in}\;\mathrm{bivalves}\;=\;\frac{{\mathrm{DA}\;\times \;d\;\times \;V\;\times \;10}^{-6}}{M}$$
where DA is the DA concentration in the diluted extract (pg mL^−1^), d is the dilution factor of the diluted extract, V is the volume of the methanolic extract (mL), and M is the mass of the bivalve sample (g).

The limit of detection (LOD) for the ELISA method was 3.3 × 10^−3^ µg DA g^−1^ wet weight for shellfish samples and 6.8 ng L^−1^ for seawater samples.

#### 5.3.4. Statistical Analyses

Statistical analyses were conducted to compare CRs and FRs as well as cDA concentrations. The normality of the distribution was verified by a Shapiro–Wilk test, and the homogeneity of the variances was verified using a Bartlett test. Variance analysis tests (ANOVA) were carried out using the “car” package and repeated measures (ANOVA) were carried out using the “lme4” package in R version 3.6.1. These tests were completed with a Tukey post-hoc test. Statistical significance was set at α = 0.05 in all tests.

## Figures and Tables

**Figure 1 toxins-13-00577-f001:**
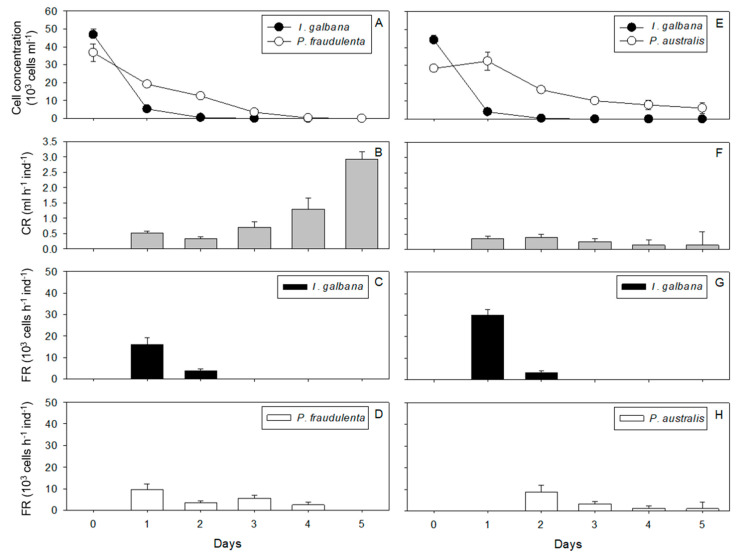
Cell concentrations (**A**,**E**), clearance rates (CRs: **B**,**F**), and filtration rates (FRs: **C**,**D**,**G**,**H**) over the 5 days of exposure of *C. gigas* to *P. fraudulenta* and *I. galbana*—condition 1 (**A**–**D**)—or *P. australis* and *I. galbana*—condition 2 (**E**–**H**).

**Figure 2 toxins-13-00577-f002:**
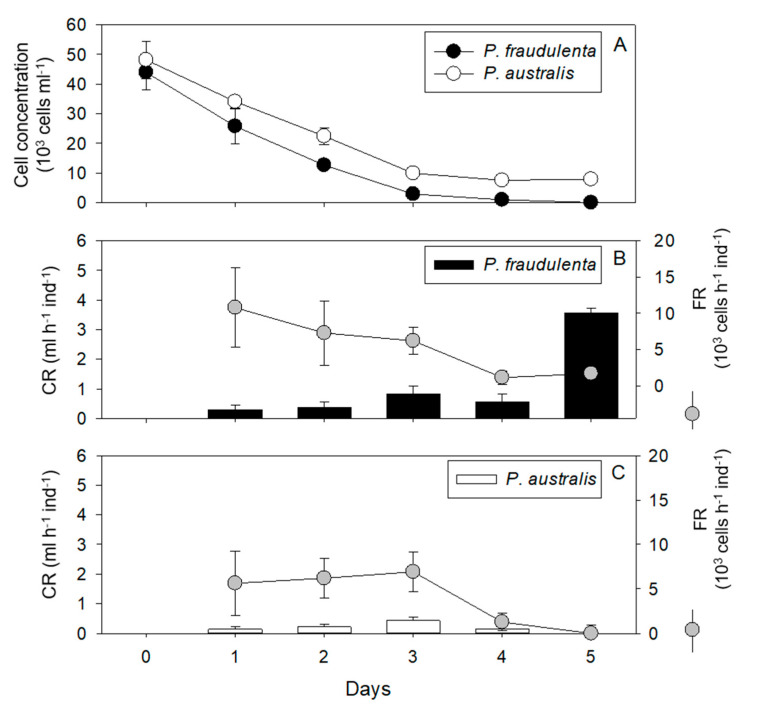
Cell concentrations (**A**), clearance rates (CRs: **B**,**C**), and filtration rates (FRs: **B**,**C**) over the 5 days of exposure of *C. gigas* to *P. fraudulenta*—condition 3 (**A**,**B**)—or *P. australis*—condition 4 (**A**,**C**).

**Figure 3 toxins-13-00577-f003:**
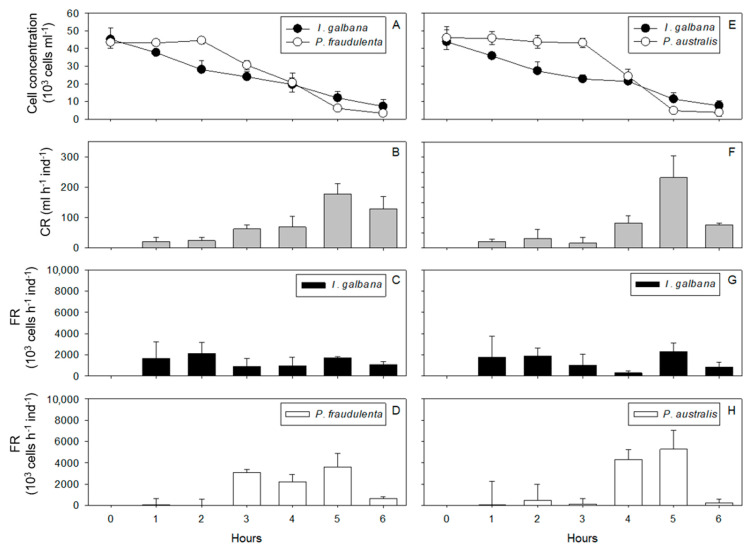
Cell concentrations (**A**,**E**), clearance rates (CRs: **B**,**F**), and filtration rates (FRs: **C**,**D**,**G**,**H**) throughout the 6 h of exposure of *P. maximus* to *P. fraudulenta* and *I. galbana*—condition 1 (**A**–**D**)—or *P. australis* and *I. galbana*—condition 2 (**E**–**H**).

**Figure 4 toxins-13-00577-f004:**
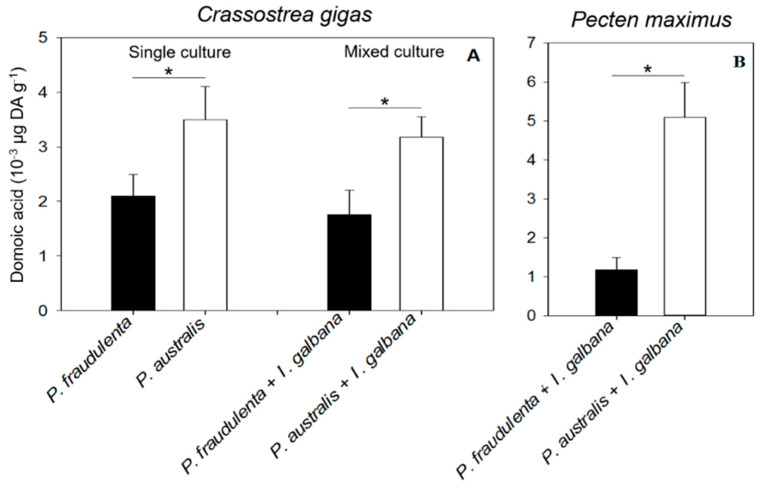
DA concentrations in *C. gigas* (**A**) and *P. maximus* (**B**) at the end of the exposure experiments. An asterisk (*) indicates a significant difference at *p* < 0.05.

**Figure 5 toxins-13-00577-f005:**
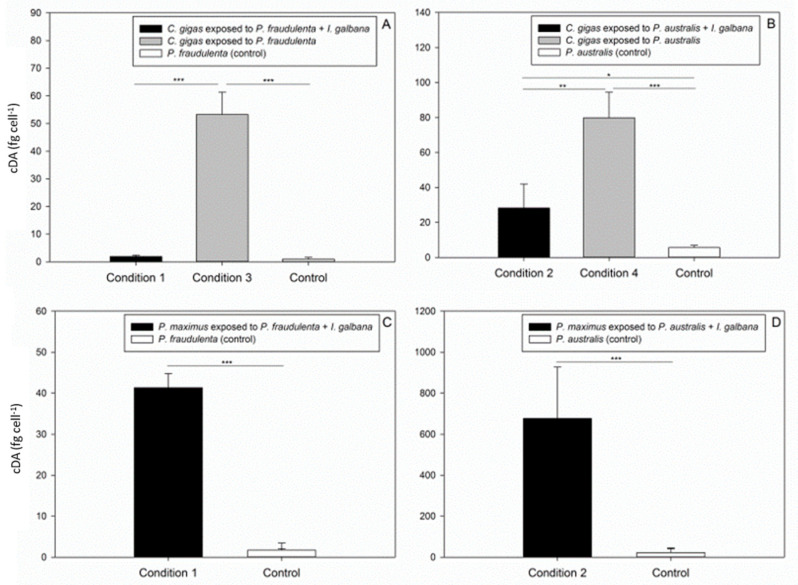
cDA concentrations (fg cell^−1^) in *P. fraudulenta* (**A**,**C**) and *P. australis* (**B**,**D**) at the end of the experiments with *C. gigas* (**A**,**C**) and *P. maximus* (**B**,**D**). The asterisks indicate significant difference at *p* < 0.05 (*), *p* < 0.01 (**) and *p* < 0.001 (***).

**Figure 6 toxins-13-00577-f006:**
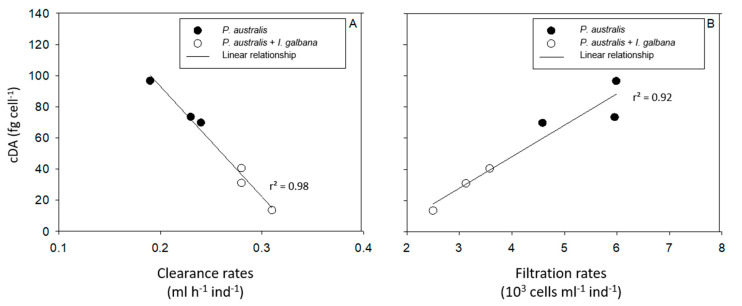
Linear regression for all data between *P. australis* cDA concentrations (fg cell^−1^) and the average clearance rates (CR, mL h^−1^ ind^−1^, **A**) or the average filtration rates (FR, cells mL^−1^ ind^−1^, **B**) of *C. gigas* throughout the 5 days of the experiment.

**Figure 7 toxins-13-00577-f007:**
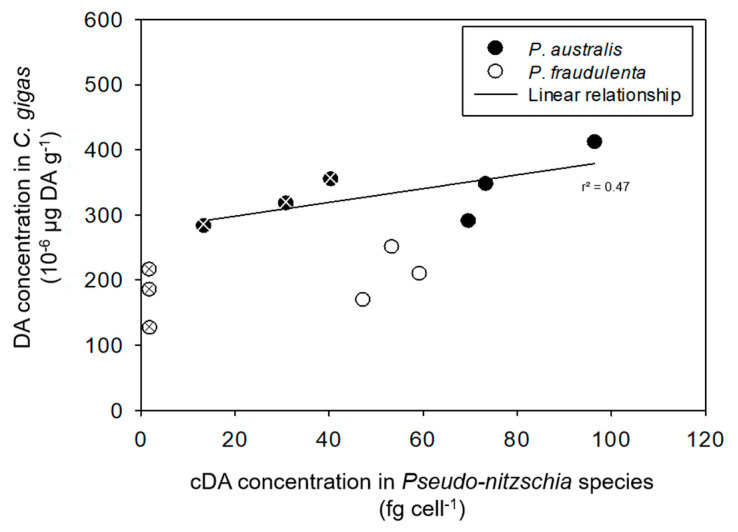
DA concentrations in *C. gigas* (10^−6^ µg DA g^−1^) as a function of cDA concentrations in *P. australis* or *P. fraudulenta* (fg cell^−1^). The linear regression only applies to *P. australis*. Crossed circles: *C. gigas* exposed to mixed cultures of *I. galbana* and *P. australis* or *P. fraudulenta*.

**Table 1 toxins-13-00577-t001:** Experimental conditions and protocols used in the four experiments with *C. gigas* and *P. maximus* exposure. Each condition was performed in triplicate.

			*C. gigas*	*P. maximus*
Experimental data		Duration	5 days	6 h
		Volume	250 mL	220 mL
		Number of individuals per condition	20 *C. gigas*	1 *P. maximus*
		Average size of individuals	1.0 ± 0.15 cm	4.10 ± 0.75 cm
		Average weight of individuals	18 ± 4 mg	1.0 ± 0.2 g
Experimental conditions	Bivalves exposed to a mixed culture of microalgae	Condition 1	38 × 10^3^ cells mL^−1^ *P. fraudulenta* 47 × 10^3^ cells mL^−1^ *I. galbana*	43 × 10^3^ cells mL^−1^ *P. fraudulenta* 45 × 10^3^ cells mL^−1^ *I. galbana*
		Condition 2	28 × 10^3^ cells mL^−1^ *P. australis* 44 × 10^3^ cells mL^−1^ *I. galbana*	46 × 10^3^ cells mL^−1^ *P. australis* 44 × 10^3^ cells mL^−1^ *I. galbana*
	Bivalves exposed to a single culture of *Pseudo-nitzschia*	Condition 3	42 × 10^3^ cells mL^−1^ *P. fraudulenta*	-
		Condition 4	48 × 10^3^ cells mL^−1^ *P. australis*	-

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
