# Peer review of "Interactions between Filter-Feeding Bivalves and Toxic Diatoms: Influence on the Feeding Behavior of *Crassostrea gigas* and *Pecten maximus* and on Toxin Production by *Pseudo-nitzschia"

_toxins, 2021, doi:10.3390/toxins13080577_

Round 1
Reviewer 1 Report
The authors present a very interesting and relevant manuscript describing interactions between two bivalve species and two toxin producing diatoms and a non-toxic haptophyte. The research is well presented and well referenced. I have only very few comments that the authors might consider.
- The species name of AC34 at Algobank-Caen is Isochrysis galbana not Tisochrysis galbana as claimed in the ms. Please correct throughout and remember to change the name also in the figures.
- line 51 and 55: I would use plural forms of brine shrimps and rotifers. Even so it was one species, it was many specimens that have been used in the experiments
- line 84, 141, and 175: C. gigas and P. maximus -> please avoid abbreviations in the headings. There is plenty of space to write the full genus names of the mussels
- line 450: that controlling control -> decide for one option. either that control or controlling
- line 453: ..., but -> change to ..., and
- line 470 and subsequently: I find the term "L:D light cycle" inconvenient. Write "light:dark (L:D) cycle" at the first occasion on line 470 and "L:D cycle" subsequently
- line 471: ventilated flasks -> was it glass or plastic culture flasks?
- line 475: you mention TEM. Why not show some TEM images in the ms that would give readers who do not encounter Pseudo-nitzschia in their daily work some better idea on how the cells actually look like
- line 483: equipped with a Nikon DS-Ri2 -> change to equipped with a Nikon DS-Ri2 camera
Author Response
Please see the attachment for the response to the reviewer's comments.

Reviewer 2 Report
Review report of the manuscript: Interactions between filter-feeding bivalves and toxic diatoms: influence on the feeding behavior of Crassostrea gigas and Pecten maximus and on toxin production by Pseudo-nitzschia
General considerations
This manuscript is aimed at clarifying some complex relationships between the production of domoic acid by planktonic diatoms and their consumption by filter feeder organisms, exemplified by the ingestion of Pseudo-nitzschia spp. by Cressostrea gigas and Pecten maximus. The ecological role of the domoic acid is largely considered in literature and its effect on filter feeder benthic organisms is known. However, the approach of this investigation is newer, in that, the reciprocal effects of producer and consumer are investigated. In fact, other researches highlighted, in the past, the effect of various factors, both biotic and abiotic, on the production of secondary metabolites by diatoms (taking into account almost exclusively planktonic consumers), but none, as far as I know, took into account the direct effect of benthic mollusks on the production of DA by diatoms. In addition, commercial bivalves may accumulate these toxins and become an issue in the case of human consumption. The language is almost correct but some minor errors were detected along the manuscript (e.g., bivalve in singular form at line 8 should be bivalves) suggesting an English linguistic revision. The manuscript is almost interesting but it would need strengthening by improving justification of the methods and explaining the actual value of some speculations. On the whole I would suggest the publication after major revision.
Abstract
Is synthetic and enough complete. However, I would suggest to add some more detailed information on the main results, at the end. Even a single sentence, after line 18, adding some more information on important results obtained. In fact, at present, the only result considered in the abstract is that “…the presence of bivalves induced an increase in the cellular DA contents of both Pseudo-nitzschia species…”.
Introduction
The introduction is well confectioned. However, it could be deepened by exemplifying some of the most interesting relationships found between DA and planktonic consumers. In addition, it would be useful to stress the importance of this investigation for the issues of human health and the possible effects of DA stored in the tissues of mollusks on human consumers. In addition:
Line 32 “DA is transferred to marine food webs”. This sentence has no common sense. DA is part of the food webs. Eventually you can state “Da is transferred to various organisms within marine food webs…”
Line 33. This may lead to confusion. I suggest to delete “by primary consumers”. In fact, its effects may become evident even if it is ingested by secondary consumers within the food webs.
Line 37. Not “dissolved form”, eventually, soluble form
Line 48 “or not”… what? Not harmful? This sentence should be revised.
Results
Results are unconventionally long in this manuscript. We understand the need to show all the results of various tests but I would suggest some synthetization, eventually by pooling some of the paragraphs and figures, in order to obtain some clearer pictures. In addition, some figures contain redundant data (e.g., not necessary to report all the initial values when they are zeroes) and the results are enriched with various discussion items, that should be avoided. I suggest to reduce drastically the results, without deleting important data for the discussion, of course.
2.1.1. and 2.1.2. In the titles I suggest to use the entire names of genera avoiding punctuations (e.g., Crassostrea gigas, not C. gigas)
“Increased significantly… decreased significantly”: you should indicate also the analysis indicating this significance, e.g., increased significantly (ANOVA, p<0.001). In any case, “p” should be in lower case.
All the figures are in a very low resolution and they are even difficult to be analyzed. I do hope the final version will contain higher resolution figures.
Line 177. Here and elsewhere you state “data not shown”. However, if this is a result of your investigations the data were found (even if they do not need to be shown in a graph). I suggest to delete this indication, since the data “are shown”, in the sense, you declare explicitly that the levels were below the detection limit. This is a datum… shown.
Line 183. This is a convoluted form. I suggest: “an asterisk indicates significant differences at p<0.05”
Line 197: add a coma after i.e.
Figure 5: the lines here appear not appropriate since you only show two times (start and end of experiments). This graph should be transformed into an histogram. Even better, since the starting values are consistently “0” I suggest to delete them all. In this case, you could propose a single plot, with four staked or paired histograms corresponding to the concentrations measured at the end of the experiment in various species. Significant differences could be conventionally indicated by asterisks
Lines 229-230. different.. different. Try to delete some annoying and confusing terms
Line 234. Again “significantly different” used without any reference here (analysis? P value?)
Lines 239-241: “Nutrient concentrations can greatly influence Pseudo-nitzschia growth and consequently DA concentrations. Therefore, we monitored the evolution of nutrient concentrations throughout the experiments.” this is a discussion or even an introductory statement, not a result.
Lines 262-264: as above
Line 277: any relationship (not “any relationships”)
Discussion
Lines 329-332 show a possible bias of the methods, where the experiments were performed on different time limits. This point should be accurately justified.
Line 366. “probably” must be better argumented to avoid speculations not sustained by collected data
Line 395: this seems the most interesting finding for this investigation, but the results should be better sustained to avoid the impression that other factors, besides the presence of the mollusks, could influence the toxin production. In addition, some doubts may be raised by the time of exposure: it is sufficient to influence the production of secondary metabolytes by diatoms? This point needs accurate discussion and justification, also based on literature support (similar or other diatoms exposed for various times to the presence/influence of consumers).
As also the conclusions state: your “findings are a first step before further studies on the feeding behavior of bivalves” and it is very important to clarify in the discussion which data are clear and well demonstrated and which data are just speculations and need further demonstration. Taking into account that the second category cannot exceed the first, to allow publication of this manuscript.
In addition, surprisingly, the quality of English in the discussion is even lower than in other parts of the manuscript. For example:
Lines 283-288. They can be deleted since they do not add anything to the introduction (or the abstract) and are not an important part of the discussion.
Line 310: “fed on”, not “fed with”
Line 310: should be “studies reported…”
Line 312: But at the beginning of the sentence should be “However, …”
Line 313: “the filtration rates of clam” should be “the filtration rates of the clam”
… and so on. Please check accurately the quality of the English text because we cannot check it all thoroughly.
Material and methods
The choice of microalgal species lefts some doubts. Apparently you have chosen diatoms based on their shape, supposing this feature may be recognized by bivalves for their prey selection. However, microalgae also have a different content in DA. So you are testing two different issues (shape and content) using a single test and this may deal to erroneous conclusions. Wouldn’t it be easier if you had chosen two algae with different shape, similar content and/or two algae with identical shape and different content? This would have not leaved doubts on the type of selection (physical or chemical?) that you discussed. You need to justify your choice here.
Line 458: used? Or maybe cultured? Tested?
Lines 540 – 545: some references to justify the formulae used might be added, unless you wrote those equations ex novo (in this case you need to justify why and how)
Line 589: “a posteriori” you probably intend “post-hoc”
Author Contributions: Funding: Institutional Review Board Statement:. Informed Consent Statement: Data Availability Statement: Acknowledgments: All these sections are totally missing and need to be completed
Author Response

(The authors gave the same response as above.)

Round 2
Reviewer 2 Report
I see that most suggestions were followed. However, some linguistic problems are still present and I would again recommend a revision by a mother language author.
Author Response
The manuscript was revised twice (before first submission and, then, after the first revision for the Discussion section, according to this reviewer comments) by a native-english, professional translator and we provided the proof for this. You will also find this proof of correction in attached file.
